# Neurobiological Features of Posttraumatic Stress Disorder (PTSD) and Their Role in Understanding Adaptive Behavior and Stress Resilience

**DOI:** 10.3390/ijerph191610258

**Published:** 2022-08-18

**Authors:** Felippe Toledo, Fraser Carson

**Affiliations:** 1LUNEX International University of Health, Exercise and Sports, 50 Avenue du Parc des Sports, L-4671 Differdange, Luxembourg; 2Luxembourg Health and Sport Sciences Research Institute ASBL, 50 Avenue du Parc des Sports, L-4671 Differdange, Luxembourg

**Keywords:** neuroanatomical research, social behavior, mental health

## Abstract

Posttraumatic stress disorder (PTSD) has been impacting the functioning of a large number of people in military activities and victims of violence for many generations. However, investments in research aiming to understand the neurobiological aspects of the disorder started relatively late, around the last third of the 20th century. The development of neuroimaging methods has greatly supported further understanding of the structural and functional changes in the re-organization processes of brains with PTSD. This helps to better explain the severity and evolution of behavioral symptoms, and opens the possibilities for identifying individual preexisting structural characteristics that could increase symptom severity and the risk of development. Here, we review the advances in neuroanatomical research on these adaptations in PTSD and discuss how those modifications in prefrontal and anterior cingulate circuitry impact the severity and development of the disorder, detaching the research from an amygdalocentric perspective. In addition, we investigate existing and contradictory evidence regarding the preexisting neurobiological features found mostly in twin studies and voxel-based morphometry (VBM) reports.

## 1. Introduction

Traumatic experiences have always been a part of life. They affect people in different ways, depending on how one perceives and processes the experience. It is pertinent to mention that not all who experience trauma will develop a psychiatric dysfunction. Exposure to heightened stress is not solely responsible for neurocircuitry alterations that affect internal thoughts and social behavior. The individual differences present before, during, and after the traumatic event play a determinant role on how this event will be processed [1]. One example of these personal differences in perception and processing is that police killings of unarmed African-American individuals seem to negatively impact the mental health of African-American spectators, but rarely lead to such responses in Caucasian Americans [2]. Stress influences brain functioning, stirring the modulation of cognitive faculties, either in a positive or protective matter by increasing the individual’s ability to mentally rebound from adversity (mental resilience), or in a negative way in the form of excessive and persistent stress that can overwhelm the compensatory system, leading to psychiatric conditions, including posttraumatic stress disorder (PTSD) [1,3,4].

Trauma-related emotional ramifications have been recognized as impacting a person’s quality of life and functioning throughout history. Originally recognized exclusively as combat-related, this collection of symptoms has carried many names, such as “soldier’s heart” (American civil war, 1861–1865), shell shock (World War I, 1914–1918), and “battle exhaustion” (World War II, 1939–1945), until it was officially denominated in 1980 in the third version of the Diagnostic and Statistical Measurement of Mental Health Disorders (DSM-3) as posttraumatic stress disorder (PTSD) [5]. Posttraumatic stress disorder affects not only the individual, but also extends its influence to the whole society of which the individual is a part. The economic impact on PTSD to health systems is also considerable, since it can affect the individual’s ability to perform their professional role in society, as well as in their family structure [3,6].

Hypervigilance, heightened sensitivity to stimuli perceived as threatening, reexperiencing phenomena, fear-conditioning, and the inability to extinguish learned fear (fear extinction) problems are particular to PTSD, being rarely observed in such synchronicity in other mental health conditions [5,7,8]. Fear conditioning is understood as the adaptive behavior responsible for dealing with life-threatening situations in a matter that the system can recognize environmental cues related to a traumatic event to avoid harm [5]. In PTSD, those adaptive responses affect functionality by being elicited even when the individual does not experience a plausible threat [5].

Posttraumatic stress disorder used to be associated with the emotional toll of war, but the condition became well associated with trauma experienced through natural disasters or abuse of any kind, which would elicit such exacerbated emotionally dysregulated responses. Around 5 to 20% of individuals exposed to traumatic events in the United States tend to develop PTSD [9,10], and, at a larger scale, a prevalence of 30 to 40% is estimated for victims of disasters, leading to about 19% of those survivors attempting suicide [11].

The processing of heightened stress, the consolidation of trauma-related memories, and the resulting inaptitude to extinguish these fear responses, causing a lack of inhibition of trauma-avoidance reminders, are linked to disorganized resting-state neural activity [12]. Neuroimaging studies have served a great deal in uncovering functional circuitry accommodations found in individuals with PTSD [3], leading to the awareness of biological markers that can be related to the social and genetic traits of the individual [1]. Identifying neurobiological markers in these networks can steer toward a deeper understanding of the condition, possible vulnerability factors, and treatment possibilities [9,12], but, even though the usual neurocircuitry model of PTSD that focuses on limbic, paralimbic, and prefrontal alterations [3] is, to date, apposite, some variability can be found in studies that investigate those markers. Different imaging, assessment, and diagnostic tools, and a great variety in the trauma modalities investigated, among other factors, have led to a lack of overall agreement in the literature [7,13].

Over the years, most PTSD and anxiety models focused on amygdalocentric circuitry for its discernible role in the fear response, conditioning, and extinction [14], with the Amygdala–Hippocampus loop (AM-HPC) receiving substantial attention for being associated with prolonged and exaggerated responses to threats observed in individuals with PTSD, where AM hyperactivity and HPC hypoactivity result in impaired ability to extinguish conditioned fear [13] during this phase of memory consolidation. Other than its subcortical connections, the AM also plays an important role in processing and recall during emotionally motivated reasoning processes occurring between different parts of the prefrontal, cingulate, and insular cortices [15], which explain many of the dysfunctional behaviors observed in people with PTSD.

The purpose of this review is to remove these amygdalocentric processes and models to take into consideration the role of the various interactions between subcortical systems on cortical regions involved in the encoding, storage, and decoding of trauma-related information that determine behavior. The focus of the analyses of structural and functional changes in the PTSD brain affecting fear extinction, inhibitory control, and, consequently, behavior should lead to the further comprehension of PTSD symptomology and possibly assist in identifying biological variations that could indicate an increased risk of developing the condition in the presence of chronic stress, as well as future prevention and treatment strategies.

Twenty-three articles were selected for this review, with a total of 10.981 participants. All papers analyzed cortical and/or subcortical structures and systems affected by exposure to prolonged trauma, either structurally or regarding activation, in participants with posttraumatic stress disorder (see Appendix A). Although a reasonable congruence in design for observational research in epidemiology can be seen in case-controlled studies, and all of the papers considered for this review were consistent in diagnosing and assessing the symptoms of participants with PTSD, a great variety of methods can still be seen in the investigation time and measurements. Most of the observational studies—as many of the studies used in the reviews encountered—applied resting-state or under-symptom-provocation voxel-based morphometry (VBM), with only two of them acquiring data through regional homogeneity (ReHo) or other methods.

Voxel-based morphometry is an automated neuroimaging method that was first presented by Ashburner and Friston in 2000 to inquire about macroscopic gray matter changes with high spatial resolution using T1-weighted MRI images, which were then statistically examined on all voxels, determining differences in the volumes of brain structures between groups [16,17]. Two studies have also applied ReHo, a neuroimaging method that allows the estimation of regional activation patterns in the encephalon based on the sum of the functional connectivity (FC) between a specific node in a network and surrounding areas [18,19]. Although there was no restriction regarding the neuroimaging method applied during the collection of studies for this paper, the discussion of the results was affected by the lower classification accuracy of ReHo in relation to VBM in comparisons between PTSD and trauma-exposed healthy controls [11].

## 2. Changes Related to Prefrontal Areas

The prefrontal cortex is associated with executive faculties, playing an overall crucial role in cognitive functions acting on decision-making processes, the processing of intrinsic and extrinsic sensory information, and behavior [20,21]. This portion of the brain can be subdivided into different sections. Among the most relevant areas for mental health and decision-making processes are probably the ventromedial PFC (vmPFC), the medial PFC (mPFC), and the dorsolateral and ventrolateral PFC (dlPFC and vlPFC). Those subdivisions were sometimes named after their specific gyri in a few of the studies used here, but for coherence, the herewith presented classification will be applied when presenting the data.

In a recent narrative review on neuroanatomical components involved in emotional regulation in PTSD, Fitzgerald and colleagues (2018) described an overall under-engagement of prefrontal areas during emotional processing in individuals with PTSD [15]. It is known that low levels of norepinephrine in the brain binding to norepinephrine alpha1 receptors can increase the activation of prefrontal neurons, while higher concentrations have an adverse effect of lowering the activation of those neurons; in other words, low levels of stress can be helpful for information processing, while high levels can be detrimental for those processes, leading to hypoactivation in the PFC [22]. This level of hypoactivation can also be observed in other disorders of the anxiety spectrum [23]. What makes it remarkable for individuals with PTSD is that, even in anxiety disorders where the degree of emotional dysregulation is easily recognized, even by non-professionals (such as social anxiety and specific phobias), the activation disturbance observed in the frontal areas in PTSD is significantly higher [8,23]. This could be caused by decreased white matter density, affecting particularly, but not solely, the superior longitudinal fasciculus [24], diminishing effectivity and the participation of logic-based systems in behavioral and decision-making mechanisms.

The ventromedial part of the PFC (vmPFC) acts on emotional processing, serving as an extensive interconnected area of the PFC associated with decision making and the perception of self [25]. In many experiments exploring the functional neuroanatomical components of mental health conditions, patients are placed under symptom provocation. It is particularly under this set that decreased activation of the left vmPFC can be seen in PTSD patients [26], affecting their ability to self-regulate negatively triggered emotions that lead to decision-making deficits and behavioral abnormalities, recognized as exaggerated reactions to environmental triggers in PTSD patients. This pattern evolves at a relatively fast pace, with the association between symptom severity in individuals with PTSD and vmPFC lower fasciculus integrity already measurable and progressing two days after exposure to a highly significant traumatic experience [27].

Due to its part in learning flexibility, extinction recall, and cue discrimination [13,28], it is evident that the vmPFC represents an important region to be investigated in individuals with PTSD. Decreased activation and reduced volume in this area can be seen during extinction learning, with the omission of activation in this phase being associated with a lack of cortical regulation over the subcortical structures of the limbic system, contributing to the aberrant behavior observed in PTSD patients under symptom-provocation settings [11,13]. This was also reflected in a study with earthquake survivors, where a negative association between GMV on the left orbitofrontal cortex and the clinical-administrated PTSD scale for the DSM-5 (CAPS-5) score was found, demonstrating the participants’ deficit in fear extinction as a lack of suppression of emotional distractors during working memory (WM) tasks [29]. Similar findings can also be seen in the case-controlled study by Ke and colleagues (2016) involving Typhoon survivors. Decreased ReHo in dlPFC, vmPFC, and AM (Figure 1) was identified in participants with PTSD when compared with trauma-exposed (TEHC) and healthy controls (HC). This finding can be linked to the exaggerated response to threats and atypical emotional regulation reported by those participants diagnosed with PTSD [3].

Further areas of high relevance for studies involving emotional regulation, fear-extinction, and vulnerability to stress are the medial, dorsomedial, and dorsolateral PFC (mPFC, dmPFC, and dlPFC) on account of their connection to the aCC and the limbic system, as well as their role in decision-making, and adaptive and social behavior. During the process of deciding upon social behavior, the mPFC and the AM operate under a mechanism of reciprocal inhibition, modulating emotional responses. Individuals with PTSD seem to present an overall deficit in this process [3,5], linked to decreased activation of the dmPFC under conflict-free conditions and during fear extinction [12,13,30], and overactivation during symptom provocation [26] when compared with traumatized and non-traumatized healthy controls.

The grounds for this issue in functional connectivity can be found in the structural changes in GMV and ReHo in the medial and dorsolateral areas of the PFC [12,31]. Diminished GMV in mPFC and around the medial frontal gyrus was found in PTSD patients (Figure 1) in studies comparing them with trauma-exposed healthy controls and non-traumatized healthy twins [12,32]. These changes in GMV may also suggest a pre-existing, inherited, or possibly acquired vulnerability to the occurrence of this imbalance when a person is exposed to severe emotional trauma [12,33].

The ventrolateral PFC (vlPFC) is another area highly involved in behavior, playing a major role in response inhibition, and influencing decision-making that affects the goal-directed response [34]. When comparing traumatized and non-traumatized healthy controls, the latter group presents lesser bilateral engagement of the vlPFC (Figure 1) during negative emotional stimulation [15]. Nonetheless, when examining the engagement of the vlPFC in similar settings, PTSD patients seem to have an even lesser engagement of this area than trauma-exposed and non-exposed healthy controls, especially in the right brain hemisphere [15,26]. This phenomenon implies that the ability to adapt behavior is highly compromised in individuals with PTSD, suppressing supratentorial activity and favoring infratentorial limbic activity on these occasions.

## 3. Modifications in the Anterior Cingulate Cortex

In view of an optimal circuitry linking the limbic system to areas responsible for decision-making, or, in other words, connecting emotional and rational processes that affect behavior, the anterior cingulate cortex plays a role in the reciprocal inhibition of the mPFC and the AM, and acts like a hub for other behavioral networks [11,35,36] responding to significant, but non-threatening, stimuli [37]. Another important feature of the aCC is the modulation of the fear response, placing it as a key structure in top-down and bottom-up processing, attention, and social behavior [38]. Due to its great functional diversity, this portion of the cingulate cortex can be subdivided into two main subregions, the pregenual aCC (paCC) and the subgenual aCC (saCC), with each playing a major role in specific subfunctions related to emotional control acting on social behavior [39]. The paCC represents a caudal part of the aCC, being mostly involved with positively affected emotional processing, while the saCC, the more rostral part of the aCC, relates mostly to processing negatively saturated emotional information [39,40,41].

Abnormalities found in the aCC–AM–IC circuitry (Figure 2) have been mostly associated with fear-based mental health conditions, while changes in the aCC–AM–PFC seem to be more common in anxiety-based disorders than other mental health issues [8,9,23,42]. Particularly on the left aCC, which connects directly to the right mPFC, dlPFC, and HPC, decreased activation can be found in participants with PTSD in response to trauma-related emotional stimulation and symptom provocation [30], establishing the importance of aCC in emotional regulation interfering with behavior in non-pathological settings.

Reduced aCC activation found in PTSD individuals compared with non-traumatized healthy controls appears to lead to a reduction in GMV within a relatively short period of time [9]. The lower white matter density and gray matter volume in the aCC found in PTSD patients [27,32] have also been discussed by some authors to be an inherited trait that increases vulnerability to the ramifications of exposure to chronic stress [30], although the evidence seems to still be weak [12,43]. In a study on earthquake survivors, a negative association was established between right aCC GMV and the CAPS-5 score, linked to personality traits of PTSD, such as harm-avoidance and alexithymia [29]. A similar set of twin studies found similar results when comparing PTSD patients with healthy controls, but could not find abnormalities in their non-traumatized twins [12].

Different studies have reported a pattern of hyperactivation of the aCC (Figure 1), especially the dorsal portion, in individuals with PTSD under symptom provocation, extinction learning, and extinction recall, as in experiments involving conditioned fear [13,26,30], while under engagement was reported during emotional processing [15]. The pattern of hyperactivation seems to be even more prominent in the aCC–IC system, and has been recognized as associated with threat detection [44], arousal of the sympathetic system, and interoceptive awareness evaluation [45], linking it to symptoms of hypervigilance and exaggerated responses to perceived threats in individuals with PTSD. During fear conditioning, this activation pattern based on morphological changes in the circuitry, akin to the inability to deactivate this system during extinction [13], leads to the chronic aspect of PTSD, with the persistence of the behavioral changes and further development of morphological aspects at this site.

In studies investigating aCC GMV in combat-exposed individuals and victims of abuse—both groups with individuals diagnosed with PTSD—reduced volume was also found, with the reduced caudal aCC volume (Figure 1) being associated with the magnitudes of the psychophysiological responses in a fear-potentiated startle paradigm [9], seen as the exaggerated physiological and behavioral responses to perceived threats in patients. The presence of diversity in the results could be related to the use of medication, which is often only partially reported, or completely omitted, in many studies. Hydrocortisol and other psychotropic pharmaceuticals have beneficial effects in reversing the aCC hypoactivation pattern in the resting state [26], creating a bias for long-term imaging studies, since it is probably the continuous diminished activation that leads to decreased GMV and WMD in these patients.

## 4. Amygdala and Related Circuits

The amygdala (AM), described officially as an anatomical structure in 1867 by Theodore Meynert, is one of the most studied subcortical arrangements in individuals with mental health disorders due to its role in emotional processing and emotional memory. This almond-shaped structure is composed of several nuclei, with the basolateral, central, and medial ones being the focus of most studies due to their connections to, among others, the vmPFC, the saCC, and other structures of the limbic system during emotional learning, emotional memory, and fear conditioning and extinction [11,46,47].

With the purpose of understanding the role of the AM in behavior, a variety of studies have been performed since its discovery, from lesion-based to pharmacological manipulation, with the results reinforcing the AM’s role in social behavior and emotionally driven learning processes [48]. Hyperactivation of the AM (Figure 1) has been associated with excessive fear learning in PTSD individuals during stimulation, fear conditioning, extinction learning, and recall [12,13,14,15,22]. These aptitudes of the AM and the difficulties that individuals with PTSD experience with such emotional and behavioral processes are most likely the result of the lack of reciprocal inhibition, as discussed above, in the circuitry between the AM, the mPFC, and the saCC [7]. The hyperactivation of the AM observed in PTSD during symptom provocation, leading to increased inhibition of the aCC and the consequent dysregulation of information processing in the mPFC, dlPFC, and vmPFC, explains the aggravated behavioral changes in this population.

Despite most studies having described a hyperactivation pattern, one case-controlled study [49] did not share these findings of AM activation during script-driven recall in PTSD patients versus trauma-exposed healthy controls. The participants of this study were diagnosed with PTSD by a specialized outpost treatment center in New Zealand following the criteria of the DSM-4, and the diagnosis was confirmed by the CAPS-5 at the time of the experiment. Although script-driven paradigm experimentation is a valid method for eliciting emotional responses in a controlled set, the use of a script as a form of emotional stimulation allows systems involved in executive control to better adapt their responses [50], possibly facilitating the top-down regulation of the amygdala by the aCC. This effect can be seen in other studies that investigated how increasing the cognitive demand of a task can modulate emotional responses by increasing PFC and decreasing AM activity [51].

There are overlaps between the symptomology of PTSD and other mental health disorders on the anxiety spectrum, which makes those intersecting symptoms relevant for analysis in experiments on people suffering from PTSD. The atrophy of the AM (Figure 1), with accompanied GMV and an increase in ReHo, can be seen in many psychiatric disorders, but there seems to be a relationship between the changes in the volume of the AM in PTSD patients and baseline anxiety levels [12,52]. The decrease in GMV in the AM was also verified in combat-exposed participants with PTSD compared with their combat-unexposed healthy twins [12], suggesting that this decrease in volume is a characteristic of chronicity in anxiety-based disorders, although a few authors still recognize the possibility of it being a preexisting, possibly inherited, anatomical trait [14]; this has not yet been confirmed, with most studies rejecting this hypothesis.

Changes in the morphology of the AM observed in individuals exposed to a chronic state of anxiety have been associated with elevated levels of fear conditioning, led by overdrawn glucocorticoid responses to stress [14,38]. Exposing the AM nuclei to this biochemical set for a longer time span might also explain the evolution of stress-related symptoms. A negative association can be found between the loss of GMV in the AM nuclei and PTSD symptom severity, with the decrease in volume clearly measurable at four to five months post-trauma exposure, becoming more significant in the 24 to 36 months that follow as symptom severity increases [53]. This makes the AM GMV an important reference for diagnosis and the control of the effect of treatment in this population, and helps to understand the progression of symptoms.

## 5. Hippocampal Circuitry and Its Relation to Behavior Modulation

The hippocampus (HPC) plays a crucial role in declarative memory and cognitive operations, including active systems of social and goal-directed behavior, through its participation in exploration and active learning by coding and storing data for time, space, facts, and events [54], as well as the conditioning and down-drive of fear responses in contextual fear processing [55]. In a similar matter, the parahippocampal gyrus (PHG) assists in the early appraisal, encoding, and automatic regulation of social behavior, constructed upon emotionally salient stimulation, through its connections to the AM and the HPC itself, as well as to ventro- and dorsomedial frontal areas [56], as described earlier by the Papez circuitry in 1937 (Figure 2). The PHG has also been associated with visuospatial processing and contextual associations, leading to emotional responses that act on regulating, or at least strongly influencing, social and goal-directed behavior [57]. The role of these two sites in the perception and interpretation of, and reaction to environmental stimuli make them clear sites of interest for any behavioral disbalance, but even more so for psychiatric conditions that disrupt adaptive and social behavior through learning paradigms, linking them to the difficulty that individuals with PTSD experience in properly contextualizing threats [3].

The responses of these areas in PTSD may depend on the environmental set and experimental conditions. Decreased activation of HPC and PHG, with decreased blood flow, can be observed in participants when not under emotional stimulation [13], while, under symptom provocation, increased bilateral activation and irrigation can be observed for both sites in individuals with PTSD [12,13,15], in association with increased activation of the left superior temporal gyrus (26). The activation of the HPC—and subsequently PHG—during symptom provocation in PTSD is most likely related to the presence of “intrusive recall of trauma-related memories” during the procedure [12], meaning that these patterns of activation would again be influenced by the top-down regulatory processes found in the script-driven set described above [50,51].

Again, the case-controlled study by Douglas and colleagues (2019) presents contradictory findings, with the authors reporting decreased blood flow to the PHG and superior temporal gyrus during script-driven neuroimaging recordings, raising the question of the validity of script-driven experimental procedures in the investigation of PTSD functional neuroanatomy and physiology. A more reliable investigative neuroimaging method in these cases might be the resting-state VBM, as seen for other regions of interest as well. Decreased HPC and PHG GMV, and decreased volume in all temporal gyri (Figure 1), have been abundantly reported in studies [7,11,12,27,31,32,33]. One of the sources of these changes is probably the interrupted neurogenesis observed in hippocampal neurons under chronic stress conditions, modifying behavior and leading to more generalized responses to threats, instead of analysis of specific events and possible hazards [22], which has been observed in exaggerated responses to triggers in individuals with PTSD, as well as in abnormal behavior in other stress-related mental health disorders.

Kasai and colleagues (2008) reported a significant volume decrease in the right HPC in a twin study, while Yamasue and colleagues (2003) reported this happening extraordinarily on the left hippocampus [33,58]. Laterality is not as widely discussed in most papers, however. Many authors do not care to report differences between right and left subcortical structures and nuclei if they do not reach significance. Beyond that, functional activation patterns can be seen in different symptoms of mental health disorders, making it difficult to specify the relevance of laterality for a specific dysfunction. In the case of the hippocampus, lesional and animal modules have shown a few discrepancies between the right and left HPC when it comes to subfunctions inside a system, with the left one being associated with declarative aspects of object allocation [59,60] and the right one appearing to be more invested in navigation [59]. Even though this difference is physiologically plausible, with the right HPC having fewer clusters of activity related to associative spatial memory, the end result for symptomology in patients struggling with mental health conditions is the same.

Despite controversies regarding the severity of the decreased hippocampal volume, there is a consensus that, in individuals with PTSD, the HPC undergoes a reduction in GMV, possibly caused by exposure to chronic stress itself (overwhelming of the system) or due to comorbid conditions often found in PTSD, such as depression and substance abuse disorder [55], which can either lead to or at least contribute to bilateral morphological adaptations in this structure.

This sign of atrophy can be seen in other mental health conditions as well, such as affective disorders and stress-related personality disorders [12,32]. This overlap might indicate congruent structural adaptations to mental health conditions, or simply be a reflection of psychiatric comorbidities. Regardless of the main reason behind the atrophy in these areas, the effect on learning (i.e., fear conditioning and extinction) is probably the major component of the diagnostic criteria for PTSD. Animal modules have determined that hippocampal lesions can appear because of prolonged stress-related toxicity, caused by increased release of glucocorticoid and glutamate, or instability in neurotrophic factors’ concentrations [7,61], leading to volatile synaptic maintenance and development around these areas.

The hippocampal volume, like other alterations described in this paper, has also been discussed as a preexisting condition that would subject the system to higher vulnerability to the detrimental effects of prolonged exposure to stress conditions. Here again, twin studies can provide important input regarding the matter. When identical twins of combat-related individuals with severe PTSD that were not exposed to combat themselves were investigated in a case-controlled study, they showed HPC volumes comparable to those of their siblings, and significantly smaller than those of veterans without PTSD and their respective siblings [55]. This could indicate that the HPC volume might be an inherited trait, acting as a predictor of the likelihood of mental health hardship in the presence of significant and chronic stress, with future implications for treatment and prevention.

## 6. Thalamus, Hypothalamus, and Their Influences on Hormonal Systems

The thalamus (THA) is an important structure of the basolateral circuitry (THA–aCC–mPFC–TP–AM) involved in emotional regulation and, therefore, indirectly involved in decision-making processes involving social behavior through its connections to other subcortical structures of the limbic system and PFC areas. Evidence suggests that emotional visual information processing [62], an important component of disorders such as PTSD, where emotional responses can be triggered by environmental cues, is at least partially allocated in the THA–AM pathways.

Comparing activation patterns on the thalamic nuclei requires the consideration of a multitude of aspects, such as possible damage or dysregulation in secondary networks, since those nuclei often act as a hub connecting and distributing information across the cerebral cortex and subcortical structures. A pattern of thalamic hypoactivation (Figure 1) is present in people with PTSD when compared with trauma-exposed healthy controls during fear conditioning, extinction learning, and recall to a greater extent than that seen in other anxiety disorders, such as specific phobias and social anxiety [8,13,26]. The same can be seen in the form of increased regional homogeneity when comparing individuals with PTSD and healthy controls [31], implying that the THA may play a role in the reciprocate inhibition between the AM and aCC, since the same configuration can be observed during fear conditioning and fear extinction under a similar set, in both the AM and the aCC [13,26,30].

In a similar way, and due to its participation in hormonal regulatory functions, the hypothalamus (HPT) has been the target of psycho-neuroendocrinology research for many years. Hyperactivity in the hypothalamic–pituitary–adrenal (HPA) axis has been observed in patients with PTSD exposed to symptom provocation, as well as in other mental health conditions, such as depression, linking it to increased systemic responses to stress [4,12]. The presence of reduced HPC volume observed in PTSD patients, as well as in animal models of fear conditioning, predispose the system to reduced neuroendocrine regulation in the HPA axis, leading to higher cortisol levels when the system is exposed to stress that causes, as a consequence, an inflated conditioned emotional response [55]. This model could be an explanation for how smaller HPC volumes may act as a preexisting vulnerability factor, or may simply be a consequence of the presence of such morphological traits.

## 7. Sensory Processing in the Insular Cortex

The posterior and midinsular cortices receive and process visceral and other sensory stimuli, transmitting this information later to the anterior portion of the insula, which will additionally process this data and link it to other regions involved in cognitive functions. The anterior insular cortex (aINS) and its connections to the aCC are crucial components of the cinguloopercular network involved in interoceptive and emotional awareness, presenting an unusually complex architecture in humans in comparison with other hominids [63]. This part of the network shows enhanced activity in the presence of personally significant stimulation, showing unfailing coactivation during exposure to emotional pain, as well as empathy for pain [64].

During conditioning, extinction learning, and extinction recall in patients with PTSD, the aINS exhibits hyperactivation [13], a pattern also observed in other structures that influence emotional processes through the aCC. Over time, a process of neurodegeneration seems to act on the integrity of the aINS, with individuals diagnosed with PTSD presenting significantly reduced GMV (Figure 1) than non-traumatized controls [32], a feature already previously observed in comparison with combat non-exposed twins [55]. The degradation of this portion of the cinguloopercular network seems to be linked to the aCC dysregulation observed in PTSD, contributing, if not leading, to the deficit in reciprocal inhibition between the aCC and the AM, starting a cascade of events that result in abnormal behavior as a response to environmental stimuli.

## 8. Role of the Basal Nuclei

Since their description in 1664 by the physician and neuroanatomist Thomas Willis, the basal nuclei (BN) have been associated with regulatory loops throughout the history of neuroscience. More recent fMRI evidence also suggests a highly important role in cognitive and emotional processes, including the evaluation of possible outcomes, learning, and decision-making processes [65]. More specifically, the accumbens nucleus (NAcc) was found to be involved in the gathering of information during emotional memory processing, having its projections to and from the HPC in operations associated with contextual fear conditioning [66].

Increased psychophysiological responses in the BN were found during symptom provocation experiments in patients with PTSD (Figure 1) when compared with trauma-exposed and non-traumatized healthy controls [12,26,31]. Even though the NAcc volume could not be found to be associated with preexisting vulnerability factors for PTSD until now, the greater GMV of this nucleus before trauma exposure seems to be related to a reduction in PTSD symptoms both pre- and post-treatment [52], showing that these patients appear to profit more from behavioral therapeutic strategies for self-regulation, such as cognitive behavioral therapy (CBT). At the same time, greater NAcc GMV, hypothesized by Burkhouse and colleagues (2020) as a ramification of the use of selective serotonin reuptake inhibitors (SSRIs), presents a negative relationship with the severity of anxiety symptoms [52,67,68]. These could possibly reflect the role of the NAcc–HPC circuit in emotional memory processing, by reinforcing symptoms of fear- and anxiety-related mental health conditions, but also by assisting in re-learning and emotional self-regulatory strategies.

## 9. Conclusions

### Final Considerations and Future Directions

Exposure to high levels of negative emotional stress leads to changes in limbic and limbic-related structures as the exposure develops into chronicity. This has been observed through processes of physiological and morphological reorganization to a perceived threat. It is unclear whether the loss of GMV in medial portions of the PFC, aCC, and aINS is narrowly a result or the sole cause of these adaptive mechanisms. Current evidence regarding limbic dysregulation affecting the HPA axis, their effects on BDNF release, and how this process may have its origins in disturbances in aCC–AM reciprocal inhibition suggests a vicious cycle. This cycle may be an explanation for the chronicity of the disorder, progressively affecting the individual’s ability to maintain social behavior stability.

The use of amygdalocentric systems to further understand symptomology is, therefore, although still highly relevant, somewhat insufficient. The collection of symptoms (such as the predisposition to a suboptimal mindset in the presence of heightened stress) seems to rely mostly on reciprocal inhibitory processes, located mainly around aCC–mPFC functioning. Traits of the PFC, for example, acting on top-down and bottom-up cingulate regulation of limbic components might be a reasonable focus for future research exploring the functioning of a vicious cycle that leads to and further develops behavioral symptoms in individuals with PTSD.

Another important facet of mental health management in society is prevention. Understanding the channels that lead to disorders affecting behavior and quality of life might be the key to reducing the occurrence, or at least the impact, of such disorders, favoring the individual and society at different levels. Reduced HPC volume and the changes in structural integrity observed in the NAcc of individuals with PTSD under successful behavioral treatment might be plausible clues to identify preexisting factors for vulnerability to stress. Further research focusing on these processes might support diagnostic precision, evaluation of treatment methods, and, to a certain level, even the prevention of stress-related mental health disorders.

## Figures and Tables

**Figure 1 ijerph-19-10258-f001:**
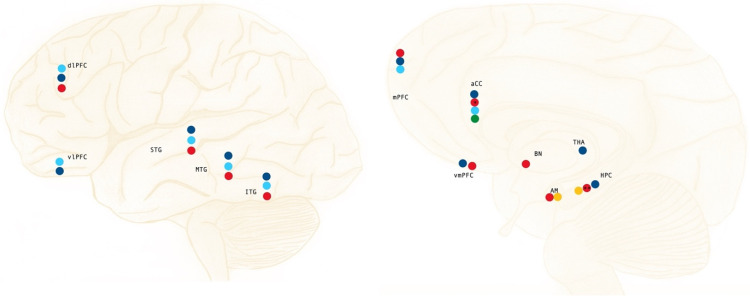
Summary of the areas and structures mostly affected by reorganization processes in PTSD; hypoactivation/under engagement under symptom provocation (dark blue), decreased white matter volume (light blue), decreased gray matter volume (red), hyperactivation during threat detection (green), and hyperactivation/over engagement during symptom provocation (orange). * Adaptation more prominent on the left-brain hemisphere; *** Contradictory evidence regarding laterality.*

**Figure 2 ijerph-19-10258-f002:**
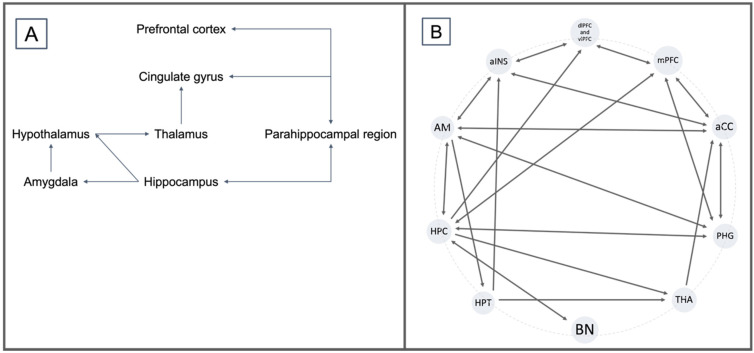
(**A**) Emotional control circuitry according to the Papez model of 1937. (**B**) Description of an off-centric model based on the Papez model and the description of neuroanatomical interactions in PTSD by Zhao et al., 2013; Aminoff et al., 2013; Zhang et al., 2016; and Benarroch et al., 2019.

## Data Availability

Not applicable.

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
