# Peer review of "Neurobiological Features of Posttraumatic Stress Disorder (PTSD) and Their Role in Understanding Adaptive Behavior and Stress Resilience"

_ijerph, 2022, doi:10.3390/ijerph191610258_

Round 1
Reviewer 1 Report
Overview and general recommendation:
The authors review the advances in neuroanatomical research about the adaptations in PTSD and discuss how those modifications in circuitry impact severity and development of the disorder, detaching the research from an amygdalocentric perspective. This is a well-written paper containing interesting and innovative results which merit publication. For the benefit of readers, clarifying a point could improve this paper.
1. The authors conclude that the use of amygdalocentric systems to further understand symptomology is somewhat insufficient and that the collection of symptoms seems to rely mostly on reciprocal inhibitory processes, located mainly around aCC - mPFC functioning. The focus on these brain regions might be better described in the abstract.
2. Although there are shown only Figure 1 and Figure 3, I found only the description of Figure 1 and Figure 2 in the draft. These points should be corrected.
3. It would benefit the reader's understanding of the papers used in Figure 3 listed in the supplementary materials for sample size, modality, and summary of results.
Reviewer 2 Report
The paper by Toledo and Carson aims to give an overview of the neurobiological features of PTSD and discuss their impact on the severity and development of the disorder. As such, it is a welcomed addition to the literature within the field as it discusses a variety of neuroanatomical features of PTSD, veering away from a pure amygdalocentric approach. The Introduction section is well written and informative. While it gives a historical perspective on PTSD as well as associated disorders, it also highlights those problems that are specific for people with PTSD. Within this section, the authors also give clear motivation behind this review, aiming to provide the readers with a more comprehensive overview and discussion of the advances in neuroanatomical research about the variety of adaptations recorded in people with PTSD.
This review included 23 articles analyzing cortical and/or subcortical structures and systems that have been affected by an exposure to prolonged trauma. The data pool and data sampling are well described, including the most common data collection methods seen in the observational studies the authors investigated. The authors also recognize, and transparently present, the limitations of the current studies within the field that could, potentially, impact the results collected in this review.
The rest of the review is well structured into logical subsections, with a clear flow of ideas. The included visualizations are a welcome addition to the current state of the art. While Fig. 1 presents a great visual overview of an off-centric model for emotional control circuitry, Fig. 3 outlines areas and structures mostly affected by the reorganization processes taking place in individuals with PTSD. These are great visual aids and, as such, bring a substantial contribution to the literature. Of note here, the Figures appear to be mislabelled. Within the text, the authors reference Fig. 1 and 2, while the visuals include what is called Fig. 1 and 3. Please check the titles of figures and references.
The literature included in the review is relevant and up to date. The concluding section of the review is clear and well-written. The authors, once again, highlight the importance of abandoning the amygdalocentric system to study PTSD and suggest pathways for further research. Thus, in view of the current work, as well as the significance of the MS, I suggest the publication of this manuscript.
